# Identification of *Fusarium* Head Blight in Winter Wheat Ears Using Continuous Wavelet Analysis

**DOI:** 10.3390/s20010020

**Published:** 2019-12-19

**Authors:** Huiqin Ma, Wenjiang Huang, Yuanshu Jing, Stefano Pignatti, Giovanni Laneve, Yingying Dong, Huichun Ye, Linyi Liu, Anting Guo, Jing Jiang

**Affiliations:** 1Collaborative Innovation Center on Forecast and Evaluation of Meteorological Disasters, Nanjing University of Information Science & Technology, Nanjing 210044, China; mahq0712@nuist.edu.cn; 2Key Laboratory of Digital Earth Science, Aerospace Information Research Institute, Chinese Academy of Sciences, Beijing 100094, China; dongyy@radi.ac.cn (Y.D.); yehc@radi.ac.cn (H.Y.); liuly35@radi.ac.cn (L.L.); guoat@aircas.ac.cn (A.G.); p17201084@stu.ahu.edu.cn (J.J.); 3Consiglio Nazionale delle Ricerche, Institute of Methodologies for Environmental Analysis (CNR, IMAA), Via del Fosso del Cavaliere, 100, 00133 Rome, Italy; stefano.pignatti@cnr.it; 4Earth Observation and Satellite Image Applications Laboratory (EOSIAL), School of Aerospace Engineering (SIA), Sapienza University of Rome, Via Salaria 851-881, 00138 Rome, Italy; giovanni.laneve@uniroma1.it; 5University of Chinese Academy of Sciences, Beijing 100049, China; 6School of Electronics and Information Engineering, Anhui University, Hefei 230601, China

**Keywords:** winter wheat, ears, *Fusarium* head blight, identification, hyperspectral, continuous wavelet analysis

## Abstract

*Fusarium* head blight in winter wheat ears produces the highly toxic mycotoxin deoxynivalenol (DON), which is a serious problem affecting human and animal health. Disease identification directly on ears is important for selective harvesting. This study aimed to investigate the spectroscopic identification of *Fusarium* head blight by applying continuous wavelet analysis (CWA) to the reflectance spectra (350 to 2500 nm) of wheat ears. First, continuous wavelet transform was used on each of the reflectance spectra and a wavelet power scalogram as a function of wavelength location and the scale of decomposition was generated. The coefficient of determination *R*^2^ between wavelet powers and the disease infestation ratio were calculated by using linear regression. The intersections of the top 5% regions ranking in descending order based on the *R*^2^ values and the statistically significant (*p*-value of *t*-test < 0.001) wavelet regions were retained as the sensitive wavelet feature regions. The wavelet powers with the highest *R*^2^ values of each sensitive region were retained as the initial wavelet features. A threshold was set for selecting the optimal wavelet features based on the coefficient of correlation *R* obtained via the correlation analysis among the initial wavelet features. The results identified six wavelet features which include (471 nm, scale 4), (696 nm, scale 1), (841 nm, scale 4), (963 nm, scale 3), (1069 nm, scale 3), and (2272 nm, scale 4). A model for identifying *Fusarium* head blight based on the six wavelet features was then established using Fisher linear discriminant analysis. The model performed well, providing an overall accuracy of 88.7% and a kappa coefficient of 0.775, suggesting that the spectral features obtained using CWA can potentially reflect the infestation of *Fusarium* head blight in winter wheat ears.

## 1. Introduction

Wheat *Fusarium* head blight (*Fusarium graminearum*) is a destructive disease in the warm and humid wheat-growing areas of the world [1]. The disease is characterized by a complete destruction of the cellular integrity of the impacted tissues, leading to cell death and degradation of chlorophyll, and the damage is mostly accompanied by a transient increase in transpiration, followed by tissue desiccation [2]. This, in turn, causes serious yield loss and quality reduction [3]. Moreover, *Fusarium* may cause serous grain contamination with mycotoxins, which are poisonous and harmful to human and animal health [4,5]. Thus, it is vital to develop a method for the identification of *Fusarium* head blight before maturity to avoid potential health risks for human and animal feed.

Some progress has been made in identifying *Fusarium* head blight in wheat ears using hyperspectral remote sensing. For instance, by using hyperspectral imagery, Bauriegel et al. [6] observed that the wavelength ranges of 500–533 nm, 560–675 nm, 682–733 nm, and 927–931 nm are very sensitive to the spectral difference between healthy and diseased ear areas under laboratory conditions, whereas the derived head blight index based on the spectral differences in the ranges of 665–675 nm and 550–560 nm are suitable for identifying *Fusarium* head blight under outdoor conditions. By applying the spectral angle mapper (SAM) method, Bauriegel et al. [2] successfully classified healthy and disease-infected ear tissues in multiple stages after inoculation based on hyperspectral and chlorophyll fluorescence imaging. Whetton et al. [7] measured yellow rust and *Fusarium* head blight in wheat and barley in four fields in Bedfordshire, UK, by employing a hyperspectral line imager (spectrograph) for online measurement. Based on hyperspectral images, Jin et al. [8] successfully classified healthy and *Fusarium*-head-blight-infected wheat using a convolutional neural network in a wild field. Relying on a push-broom hyperspectral imaging system in the visible-near-infrared (Vis-NIR) range, Zhang et al. [9] proposed a specific *Fusarium* head blight classification index based on the band combination of 417, 539, and 668 nm for detecting diseased winter wheat spikelets. Its identification accuracy increased by 30% compared with that of the best-performing commonly used spectral vegetation index. Huang et al. [10] evaluated the ability of the spectral features of first-order derivatives, the spectral absorption features of the continuum removal, and vegetation indices to identify *Fusarium* head blight of wheat ears from the Analytical Spectral Devices (ASD) spectrometer. These studies mainly explored the performance of the spectral signal in the wavelength range 400–1000 nm for identifying wheat *Fusarium* head blight. Furthermore, a few scholars have also explored the response of spectral signal in the short-wave infrared band (SWIR, greater than 1000 nm) to wheat *Fusarium* head blight. For instance, Mahlein et al. [11] found that, from 12 days after inoculation (dai) onwards, the spectral signal of wheat spikelets infected by *Fusarium* head blight changed considerably in comparison with the non-inoculated control parallel to the development of infestation, that is, higher reflectance in the visible and SWIR regions and lower reflectance in the NIR region was pronounced. Alisaac et al. [12] found the high correlation between wheat *Fusarium* head blight and the spectral signal in wavelength ranges of 430–525 nm, 560–710 nm, and 1115–2500 nm. Meanwhile, their results illustrated that the classification accuracy of healthy wheat and *Fusarium*-head-blight-infected wheat using the whole spectral reflectance which considered the water stresses detected in the SWIR region caused by the disease was higher than the spectral vegetation indices from 8 dai onwards. Additionally, Dammer et al. [13] developed a color (include red, green, and blue bands) and a multispectral (include red and infrared bands) camera system with real-time image analysis software for the detection of *Fusarium* head blight symptoms. Their results suggest that remote sensing can be an effective technique for nondestructively identifying *Fusarium* head blight.

Continuous wavelet analysis (CWA), as an emerging spectral analysis method, has been employed to detect and discriminate crop diseases and pests. For instance, Zhang et al. [14] accurately estimated the disease severity of powdery mildew on leaf level through the combination of CWA and partial least square regression. Zhang et al. [15] and Shi et al. [16] revealed the promising potential of CWA for the identification of wheat yellow rust. By using CWA, Luo et al. [17] quantified wheat aphid infestation successfully. In quantifying crop diseases, wavelet features were demonstrated to outperform conventional spectral features [18,19]. Additionally, some studies illustrated that CWA performed well in differentiating crop stresses [20,21,22,23,24]. The above results demonstrate the superiority of CWA for crop pest and disease monitoring. However, for the identification and detection of *Fusarium* head blight in wheat ears using hyperspectral data, the current studies are mainly based on spectral processing methods such as the SAM method [2], principal component analysis [25], optimal bands selection using exhaustive searches in NIR and visible bands [26], disease index construction using specific bands [6,9], in-field visual assessment, and photo interpretation assessment [7]. The identification of wheat *Fusarium* head blight based on CWA has not been reported yet. Therefore, although the wavelet features obtained via CWA have been widely used for the identification and discrimination of crop pests and diseases, the performance of wavelet features for the identification of *Fusarium* head blight in wheat still remains unclear and it should be further explored.

Fisher linear discriminant analysis (FLDA) [27,28] is a kind of classic and popular supervised learning method. It attempts to find a linear transformation that maximizes the dispersion between classes and minimizes the dispersion within the class to separate one class from the others [29]. FLDA is commonly used in recognition, classification and feature extraction [10,30,31,32,33]. The existing successful cases support the use of FLDA in this study for the identification of *Fusarium* head blight in wheat ears.

In this study, an FLDA identification model based on the wavelet feature set extracted using CWA was developed for identifying *Fusarium* head blight in winter wheat ears. Two independent hyperspectral experiment datasets in the range 350–2500 nm obtained during the wheat filling stages in 2018 and 2019 were used. This study aimed: (1) to evaluate the efficiency of CWA for identifying *Fusarium* head blight; (2) to determine the most informative wavelet features for identifying *Fusarium* head blight.

## 2. Materials and Methods

### 2.1. Experimental Areas

Two experiments were conducted in this study.

Experimental 1 (Exp. 1): The *Fusarium* head blight experiment for winter wheat ears was conducted at three experimental fields at the wheat grain filling stage from 26 April to 9 May, 2018. These three fields are respectively in Taoxi Town (experimental field 1, 31°32′N, 116°59′E) in Shucheng County, Lu’an City, Guohe Town (experimental field 2, 31°29′N, 117°13′E), and Baihu Town (experimental field 3, 31°14′N, 117°27′E) in Lujiang County, Hefei City, Anhui Province, China.

Experimental 2 (Exp. 2): Combing with the actual occurrence status of the disease in the three experimental fields in 2019 (the disease occurred only in experimental field 2), the *Fusarium* head blight experiment for winter wheat ears only continued at experimental field 2 in Guohe Town in Lujiang County, Hefei City, Anhui Province, China at the wheat filling stage from 2 to 10 May, 2019.

All the above experimental regions are located in Anhui Province, which belongs to the transitional climate zone between warm temperate and subtropical zones. The average annual temperature of the province ranges from 14 to 17 °C and its annual precipitation ranges from 700 to 1700 mm [34]. *Fusarium* head blight occurs frequently here [35]. Additionally, *Fusarium* oxysporum is abundant in the experimental regions, and there is grave occurrence of *Fusarium* head blight in wheat. Furthermore, the experimental regions are prone to rainy weather during the wheat heading and flowering period, and a high-temperature and high-humidity environment is easily formed, which is conducive to the occurrence of *Fusarium* head blight in wheat [36]. Thus, within the two years of the experiments, *Fusarium* head blight in the experimental regions occurred under natural conditions without the need for manual intervention such as inoculation. On the other hand, according to the local meteorological data from Anhui Meteorology Service (http://ah.cma.gov.cn/), excessive precipitation and high temperatures occurred during the critical period of *Fusarium* head blight infestation in 2018, which was conducive to the *Fusarium* head blight epidemic. However, in the same period of 2019, not only insufficient precipitation but also continuous low temperature occurred, which was not conducive to the disease infestation. The above results may be the main reasons behind the difference in the disease incidence among the three experimental fields over the two years.

### 2.2. Data Acquisition

#### 2.2.1. Wheat Ear Spectra Measurement

In practice, as wheat *Fusarium* head blight may infest any part of the ears, and the reflection of the disease infection status on each side of one ear might be different, the spectral information on each side of the ear hence needs to be collected to capture the information on disease infection. However, the conventional spectrum measurement technologies are perpendicular to the crop canopy during spectrum acquisition, i.e., the collected spectrum mainly reflects the information of the top of the wheat ears. Therefore, in this study, in order to collect much more effective spectral information of the wheat ears to reflect the disease infection status, wheat plants were cut from the field with scissors, and the ears were fixed to the center of a 1 m × 1 m black cloth and then immediately spectrally measured to obtain the group spectral of two different sides of the ears (Figure 1). When measuring the spectrum of one side (side 1 or side 2) of the ear, the corresponding side of the ear was placed upward and the ear was then fixed on the black cloth with double-sided tape, which did not affect the spectrum. To prevent the change of the spectrum of the cut plants over time, all spectral measurements were done in the field. Hence, the black cloth was used as a background to separate the ear from other objects in the experimental field to ensure that was the only one on the field of view of the spectroradiometer.

The hyperspectral data of wheat ears were measured using ASD FieldSpec Pro spectrometer (Analytical Spectral Devices, Inc., Boulder, CO, USA) in an open field at the wheat grain filling stage in 2018 and 2019. The spectroradiometer was fitted with a field of view of 25°. All ear spectral measurements were taken at a height of 0.5 m above the black cloth. The spectral range of the spectrometer was from 350 to 2500 nm, with the spectral resolutions of 3 and 10 nm in the 350 to 1000 nm and 1000 to 2500 nm regions, respectively. A 40 cm × 40 cm BaSO_4_ calibration panel was used at every 10 measurements to correct the changes in the illumination condition. All the experiments were conducted under cloudless conditions between 10:00 a.m. and 14:00 p.m. (local time) when minimum variations in solar view angle occurred. The reflectance spectrum of each side of each ear was measured 10 times, and the average of these measurements was considered as the reflectance spectrum of one side of the ear. The average of all the different sides of the ear was considered as the reflectance spectrum of the sample. In Exp. 1, the spectral reflectance of 87 winter wheat ear samples was collected, in which 20 ear samples were from experimental field 1, 39 ear samples were from experimental field 2, and 28 ear samples were from experimental field 3. Using the same method, the spectral reflectance of 127 winter wheat ear samples was collected in Exp. 2.

#### 2.2.2. Determination of Disease Infestation Ratio (DIR)

The DIR of all sampling ears was inspected according to the National Rules for Monitoring and Forecast of the Wheat Head Blight (Fusarium graminearum Schw./Gibberella zeae (Schw.) Petch), issued in 2011 (GB/T 15796–2011). The number of the disease infested spikelets and all spikelets (including both healthy and infected) of each ear was firstly counted by visual interpretation. The DIR of each ear was then determined by the ratio of *Fusarium* head blight damaged spikelets among all the spikelets of the ear and its value was calculated with in a range of 0% to 100%. Where 0% represents healthy and 100% represents the severest disease infection. In this study, the disease infestation conditions per ear were reorganized into two classes for subsequent identification analysis: healthy (infection ratio: ≤10%) and *Fusarium* head blight infected (infection ratio: >10%). Ears with an infestation ratio ≤10% were assigned to the healthy class due to the difficulty to accurately separate them from healthy ones. Table 1 summarizes the disease field investigation experiments of the two years.

### 2.3. Preprocessing and Standardization of Spectral Reflectance Data

Owing to the influence of the high relative humidity of the air in the study area, severe noise always occur at the spectral wavelength ranges of 1350 to 1420 nm, 1800 to 2000 nm, and 2350 to 2500 nm in the short-wave infrared (SWIR) region [37,38] (Figure 2a). Therefore, these three ranges were removed for subsequent analysis in this study. Both Exp. 1 and Exp. 2 were strictly conducted according to Section 2.2.1; hence, the difference of the background information such as crop growth condition and measurement environment of the two-year experiments was regarded as the key factor influencing the spectral measurements. To reduce these differences over the two years, the data of Exp. 2 were adapted to match the data of Exp. 1 by dividing a ratio spectral curve. The ratio was a result of the averaged spectral curve from the healthy samples in Exp. 1 divided by the averaged curves from the healthy samples in Exp. 2. Figure 2b shows the produced ratio curve. Consequently, the spectral data collected in Exp. 2 were divided by the corresponding ratio curve to generate a set of standard spectra which were close to the level of the data in Exp. 1. This increases the comparability between the datasets of the two experiments by eliminating the possible spectral difference caused by the background information between the two experiments, without changing the inner relationship reflected by the original data.

### 2.4. Wavelet Features Extraction for Fusarium Head Blight Using CWA

The feasibility of CWA for the hyperspectral-data-based identification and detection of crop pests and diseases has been demonstrated [14,15,16,17,20]. Continuous wavelet transform (CWT) [39] is a wavelet analysis method for localizing the signal simultaneously in the time-frequency domain to detect and analyze weak signals at various scales and resolutions and to analyze multidimensional hyperspectral signals across a scale continuum [14,16,40]. Based on CWT, the original reflectance spectrum of each *Fusarium*-head-blight-infected ear is first converted to a wavelet coefficient spectrum set on multiple scales with a mother wavelet function, in which each scale corresponds to a frequency of spectral change: high scale corresponds to low frequency and low scale corresponds to high frequency. Each wavelet coefficient spectrum has the same number of bands as the original reflectance spectrum. Furthermore, low-scale wavelets will capture the narrow absorption features of the original spectrum and high-scale wavelets will capture the continuum shape [41,42]. The output of CWT in the transformation process is given as follows [43]:(1)$$W_{f}(a,b)=\int_{-\infty }^{+\infty }f(\lambda )\;\psi_{a,b}(\lambda )d\lambda ,$$
where *f*(*λ*) is the original spectrum, λ = 1, 2, …, *m*, *m* is the number of bands, and herein *m* is 2151. *W_f_* (*a*,*b*) represents the wavelet coefficients that constitute a scalogram. $\psi_{a,b}(\lambda )$ represents the mother wavelet function of wavelet transformation as follows:(2)$$\psi_{a,b}(\lambda )=\frac{1}{\sqrt{a}}\psi (\frac{\lambda -b}{a}),$$
where *a* is the scaling factor representing the width of the wavelet and *b* is the shifting factor determining the position of the wavelet.

The Mexican hat wavelet, which is similar to the vegetation absorption characteristics, was used as the mother wavelet base in this study [39,44]. To reduce the complexity whilst ensuring the precision, only the wavelet powers at dyadic scales i.e., 2*^n^* (*n* = 1, 2, …, 10) were used [40].

Based on a threshold method, the wavelet features were finally extracted from the correlation scalogram. To find the most informative wavelet features for *Fusarium* head blight, the filtrate principles combing linear regression, independent *t*-test, and correlation analysis were applied as follows:Determine the sensitive wavelet regions with *Fusarium* head blight. A linear correlation analysis is first performed to determine the coefficient of determination (*R*^2^) between wavelet features and the DIR. The coefficients of determination (*R*^2^) between wavelet coefficients and DIR were generated to relate the scalogram with the disease infestation of wheat ears [14,39]. The top 5% ranking in descending order based on the *R*^2^ values of the correlation scalograms will be considered as the preliminary selection of wavelet regions with *Fusarium* head blight in this study. In addition, the *p*-value of independent *t*-test [45] can indicate the significance level of the difference between healthy and disease-infected eras. Thus, the statistically significant (*p*-value of *t*-test < 0.001) wavelet regions among the top 5% regions will be retained as the final sensitive wavelet feature regions.Determine the preliminary wavelet features. To reduce the computational load, only the features with the highest *R*^2^ within each wavelet region are retained as the preliminary wavelet features.Identify the optimal wavelet features for identifying *Fusarium* head blight. To reduce the redundancy among the wavelet features further, the coefficient of correlation *R* among the preliminary features will be calculated and summarized. The larger the absolute value of *R*, the greater is the correlation between the two wavelet features, i.e., the greater is the redundancy [46]. In this study, we assumed that only those preliminary wavelet features with an absolute *R* value lower than 0.8 are considered to have both strong correlation and relatively low redundancy. For the two mutually redundant preliminary wavelet features, the one with the higher correlation with the DIR will eventually be retained as the optimal wavelet features for the identification of *Fusarium* head blight.

All CWA-related analyses were performed in MATLAB 2016a software.

### 2.5. Testing the Performance of the Wavelet Features for Fusarium Head Blight

In this study, the FLDA algorithm was used for testing and comparing the performance of the wavelet features for the identification of *Fusarium* head blight. FLDA constructed a classification model using a k-means clustering based non-parametric method [47]. For a total of 214 ear samples collected from the two experiments (both Exp. 1 and Exp. 2), two-thirds of the samples in each experimental, a total 143 ear samples, were randomly selected for identification model training, and the remaining one-third of the samples, a total 71 ear samples, were used for validation. The identification model based on the wavelet features was then constructed using FLDA to evaluate the efficiency of the wavelet features extracted through CWA for the identification of *Fusarium* head blight in winter wheat ears. A confusion matrix was used to describe these assessments. Specifically, the producer’s accuracy (PA), user’s accuracy (UA), overall accuracy (OA) and kappa coefficient were calculated to assess the FLDA model from different aspects. FLDA was implemented using SPSS 22.0 software (IBM Corporation, New York, NY, USA).

## 3. Results and Discussion

### 3.1. Changes in Reflectance Spectral Owing to Fusarium Head Blight

Figure 3 illustrates the curves of the average original spectral reflectance, the reflectance ratios of healthy and *Fusarium*-head-blight-infected wheat ears, and the coefficient of correlation (*R*) and determination (*R*^2^) between the spectral reflectance and the DIR. By comparing the spectral differences between *Fusarium*-head-blight-infected and healthy wheat ears, it can be observed that the spectral reflectance of the disease-infected wheat ears gradually increased in the wavelength range of 350 to 517 nm and 580 nm to 716 nm, which is mainly in the visible region, and the wavelength range of 1162 to 2350 nm (except the wavelength ranges of 1350 to 1420 nm and 1800 to 2000 nm) in the SWIR region, whereas the spectral reflectance change was not evident in the visible range from 518 to 579 nm and the NIR range from 717 to 1161 nm (Figure 3). The initial infection symptoms of *Fusarium* head blight appear as small, water-soaked brownish spots at the base or middle of the glume, or on the rachis [48]. Water soaking and discoloration then spread in all directions from the point of infection, and a salmon-pink to red fungal growth may be observed along the edge of the glumes or at the base of the spikelet [49]. Infected grains shrink become grey/brown with a floury discolored interior. Premature death or bleaching of the spikelet is also a common symptom, giving a “scabbed” appearance. When wheat ears are severely affected by *Fusarium* head blight, the peduncle may turn dark brown [50,51]. Spectral changes during disease development are based on variations in the content of carotenoids and chlorophylls, resulting in the above infection symptoms [6]. Normally, the infected ears have a relatively lower water content than the healthy ears, which would cause the *Fusarium* head blight symptoms on wheat ears and the changes of ear spectral reflectance [11,52]. Additionally, the significant cellular changes occur after the mycelia penetrates the kernels, such as denaturation of the cytoplasm and organelles, decomposition of the host cell wall, and deposition of material in the vessel wall of the infected ears, and the damage is mostly accompanied by a transient increase in transpiration and tissue desiccation [2,53]. Therefore, the diseased infected ears exhibit a higher reflectance in the visible and SWIR regions than healthy ears. Moreover, linear correlation analysis was used to undertake an initial pass on the wavebands to evaluate the significant relationships between the spectral reflectance and the DIR in wheat ears. The result showed a significant correlation in the visible and SWIR regions, which illustrated the potential of hyperspectral data for identifying *Fusarium* head blight (Figure 3c).

### 3.2. Wavelet Features and Their Sensitivities to Fusarium Head Blight

Based on CWA, a correlation scalogram for DIR and the spectral reflectance of wheat ears is generated in Figure 4. The *R*^2^ values obtained for the correlation calculated between wavelet power and DIR were reported using the correlation scalogram at each wavelength and scale. The *R*^2^ values in this case study range from 0 to 0.602. The wavelet feature selection first retained the features where the independent *t*-test was significant (*p* < 0.001) among the top 5% features (the threshold *R*^2^ value was 0.337), which were ranked in descending order based on the *R*^2^ values. Thus, we considered the obtained wavelet feature regions (highlighted in orange color in Figure 4) to be sensitive to *Fusarium* head blight. The informative wavelet feature regions were mainly concentrated in the visible and SWIR regions, which are consistent with the sensitive regions of the original spectrum (Figure 3). Furthermore, in contrast to the original spectrum, some of the sensitive wavelet feature regions were in the NIR region, which indicated that the spectral sensitivity of some wavelengths was enhanced by CWT. 

The features with the highest *R*^2^ were retained in each feature region, which yielded 21 preliminary wavelet features. Then, the coefficient of correlation *R* between every two preliminary wavelet features was calculated to evaluate the redundancy of the features, and the *R* values of correlation between every two wavelet features of all 21 preliminary wavelet features are illustrated in Table 2. Through threshold screening (*R* value < 0.8), six wavelet features were finally identified, and their wavelengths and scales are summarized in Table 3. The six wavelet features were all identified at low scales (1 ~ 4). Four wavelet features (471 nm, scale 4), (696 nm, scale 1), (841 nm, scale 4) and (963 nm, scale 3) captured the narrow absorption features, which were primarily influenced by pigment concentration such as chlorophylls and carotenoid, and the two remaining wavelet features (1069 nm, scale 3), and (2272 nm, scale 4) captured broad changes in water content and the internal cellular structure. Furthermore, the strongest relationship between *Fusarium* head blight of ears and wavelet power for an individual feature was located in the red edge in scale 1 and at 696 nm with the *R*^2^ of 0.602. Thus, for an individual wavelet feature, the red edge is the most sensitive to *Fusarium* head blight.

### 3.3. Capabilities of the Wavelet Features to Identify Fusarium Head Blight

The identification model using FLDA is then fitted to wavelet features to evaluate the effectiveness of identifying *Fusarium* head blight. Table 4 summarizes the identification ability of the wavelet features for *Fusarium* head blight in winter wheat ears. The overall accuracy was 88.7% and the kappa coefficient was 0.775. As for healthy and *Fusarium*-head-blight-infected ears, the wavelet features produced a PA of 86.1% and 91.4%, and a UA of 91.2% and 86.5%, respectively. Furthermore, to evaluate the contribution of the wavelet features in the SWIR region for the identification of the wheat *Fusarium* head blight, the FLDA identification model based on the four wavelet features concentrated in the spectral wavelength range of 400 to 1000 nm was also constructed. The results revealed that the overall accuracy of the model using the wavelet feature set concentrated in the range of 400 to 1000 nm decreased by 2.8% more than the model using the wavelet feature set, which considered the spectral changes in the SWIR region caused by *Fusarium* head blight (Table 4). The above results illustrated that the wavelet features performed well in identifying healthy and *Fusarium*-head-blight-infected wheat ears, which indicated the feasibility of CWA for identifying *Fusarium* head blight in winter wheat. Considering the spectral change between the healthy ears and the *Fusarium*-head-blight-infected ears detected in the SWIR region was beneficial to improve the disease identification precision.

### 3.4. Implications under Field Conditions

Our results demonstrate the promising potential of CWA for identifying *Fusarium* head blight in wheat ears, which is the first step in using remote sensing technology to identify *Fusarium* head blight in a wheat field. Based on the result, wavelet features can be used for further exploration of suitable spectral features for identifying *Fusarium* head blight using field hyperspectral data, and specific sensors based on these efficient wavelengths or spectral features for practical use may be developed in the future.

Furthermore, owing to the particularity of the *Fusarium* head blight, the pathogen mainly damages wheat ears and may be randomly distributed in any part of the ears. The conventional spectrum measurement techniques, which are perpendicular to the crop canopy may result in a large loss of the favorable spectral information for the identification of *Fusarium* head blight. Therefore, this study measured the spectral from the side of the infested ears to capture information of the disease infection as much as possible, and our good identification results of the wheat *Fusarium* head blight have demonstrated the feasibility of this spectral measurement method (Table 4). By relying on a specific large machine such as a tractor or a tool carrier, the existing studies have also successfully applied this method of measuring spectral from the ear side for the identification of *Fusarium* head blight in the field [7,13]. However, if comprehensive and continuous disease detection is required to be performed, that is, the crop spectral information in the whole field should be collected, then the spectral measurement method relying on the above field equipment may be complex and may cause some damage to crops. Recently, with the rapid development of unmanned aerial vehicle (UAV) technology, crop spectral collection can be completed at multiple angles [54,55]. Compared with some field spectral measurement equipment (such as tractors and tool carriers), UAV may not only be fast and effective, but also non-destructive in the collection of spectral information throughout the whole field crop. Additionally, although the collected wheat ear samples contained multiple wheat varieties, the possible influence of the different varieties on the identification of wheat *Fusarium* head blight was not considered in this study. Therefore, the combination of multi-angle UAV technology and CWA might also be applicable for the identification of *Fusarium* head blight in the field in the future. The influence of the different wheat varieties on the performance of the disease identification model should also be explored in the future.

## 4. Conclusions

Based on two groups of independent hyperspectral measurements, this study explored the possibility of identifying *Fusarium* head blight in winter wheat ears through the spectral response features of ear infected by *Fusarium* head blight. The study demonstrated the feasibility of applying CWA to ear spectral reflectance for identifying *Fusarium* head blight. Using CWT, the reflectance spectra could be decomposed into various scales, which can effectively identify meaningful spectral information relevant to *Fusarium* head blight. Six wavelet features for *Fusarium* head blight were identified based on the combinations of a threshold *R*^2^ value (*R*^2^ = 0.337), independent *t*-test, and correlation analysis to reduce redundancy between wavelet features. The features were related to pigment content, cellular structure, and water content. An FLDA model was constructed based on the six wavelet features. An overall accuracy of 88.7% and a kappa coefficient of 0.775 were obtained, confirming the effective performance of the wavelet features for identifying *Fusarium* head blight in winter wheat ears. In the future, we will focus on testing the feasibility of CWA combining different spectral acquisition angles to identify *Fusarium* head blight in winter wheat at field scales using UAV hyperspectral technology.

## Figures and Tables

**Figure 1 sensors-20-00020-f001:**
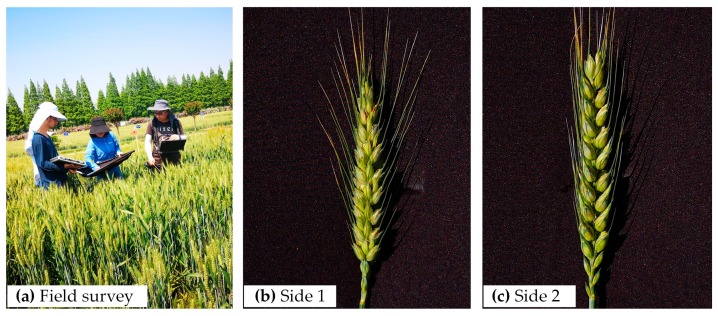
Field survey and two different measuring sides of each ear infected by *Fusarium* head blight.

**Figure 2 sensors-20-00020-f002:**
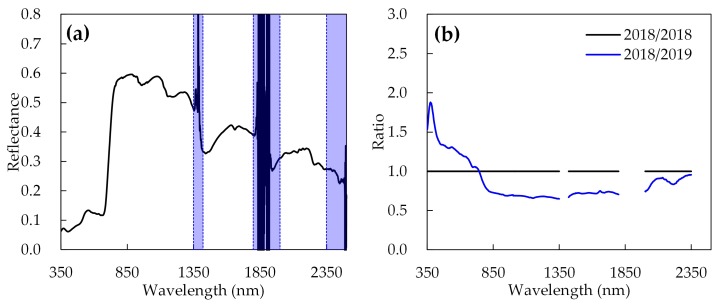
(**a**) Average spectral reflectance curve of all ears in Exp. 1; (**b**) ratio curve for data standardization between two different years.

**Figure 3 sensors-20-00020-f003:**
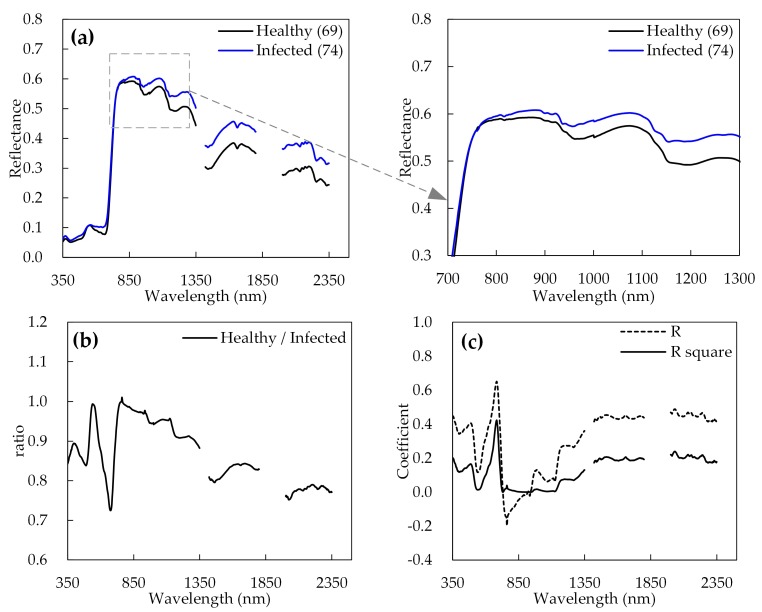
(**a**) Average spectral reflectance of healthy and *Fusarium*-head-blight-infected wheat ears; (**b**) spectral ratios of the *Fusarium*-head-blight-infected wheat ears compared with those of healthy wheat ears; (**c**) correlation coefficient *R* and determination coefficient *R*^2^ between DIR and the spectral reflectance of infected ears.

**Figure 4 sensors-20-00020-f004:**
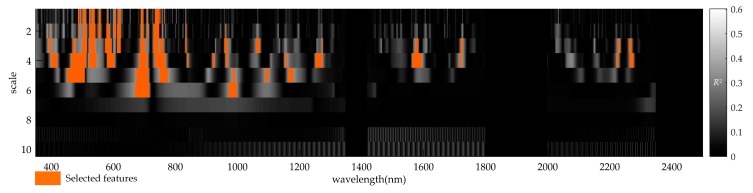
Visualization of correlation scalograms of CWA produced with the *Fusarium* head blight dataset. The selected regions highlighted orange encompass the features with the *R*^2^ values among the top 5% and which are statistically significant (*p*-value < 0.001) of independent *t*-test.

**Table 1 sensors-20-00020-t001:** Basic information for the disease survey experiments.

Experiments	Number of Field Survey Ears		
	Healthy	*Fusarium* Head Blight Infected	Sum
Exp. 1 (2018)	34	53	87
Exp. 2 (2019)	71	56	127

**Table 2 sensors-20-00020-t002:** Summary of the 21 preliminary wavelet features selected from the intersection of correlation scalograms for the disease identification.

Wavelet Features	Correlation Coefficient among Different Wavelet Features																				
	WF01	WF02	WF03	WF04	WF05	WF06	WF07	WF08	WF09	WF10	WF11	WF12	WF13	WF14	WF15	WF16	WF17	WF18	WF19	WF20	WF21
WF01	1.000																				
WF02	0.648	1.000																			
WF03	0.507	0.737	1.000																		
WF04	0.589	0.811	0.797	1.000																	
WF05	0.534	0.853	0.824	0.841	1.000																
WF06	0.715	0.736	0.734	0.866	0.824	1.000															
WF07	0.606	0.739	0.711	0.849	0.844	0.940	1.000														
WF08	0.530	0.694	0.844	0.714	0.779	0.704	0.706	1.000													
WF09	0.435	0.635	0.773	0.643	0.703	0.578	0.553	0.913	1.000												
WF10	0.389	0.763	0.766	0.729	0.812	0.650	0.659	0.807	0.702	1.000											
WF11	0.491	0.799	0.844	0.796	0.851	0.727	0.731	0.871	0.747	0.970	1.000										
WF12	0.432	0.768	0.744	0.711	0.813	0.636	0.644	0.777	0.657	0.931	0.919	1.000									
WF13	0.467	0.669	0.700	0.679	0.680	0.674	0.699	0.680	0.547	0.714	0.786	0.700	1.000								
WF14	0.364	0.734	0.738	0.695	0.755	0.562	0.589	0.753	0.616	0.943	0.938	0.913	0.723	1.000							
WF15	0.373	0.759	0.748	0.714	0.784	0.581	0.604	0.766	0.650	0.960	0.947	0.910	0.709	0.987	1.000						
WF16	0.386	0.760	0.754	0.720	0.783	0.594	0.624	0.762	0.625	0.943	0.946	0.909	0.738	0.997	0.990	1.000					
WF17	0.589	0.774	0.731	0.688	0.777	0.741	0.681	0.751	0.640	0.862	0.880	0.852	0.756	0.847	0.835	0.855	1.000				
WF18	0.654	0.900	0.695	0.766	0.820	0.814	0.806	0.658	0.579	0.680	0.727	0.700	0.689	0.639	0.667	0.677	0.788	1.000			
WF19	0.536	0.827	0.689	0.700	0.710	0.554	0.548	0.665	0.636	0.710	0.750	0.695	0.646	0.733	0.757	0.751	0.693	0.757	1.000		
WF20	0.796	0.704	0.572	0.511	0.600	0.612	0.519	0.645	0.615	0.549	0.607	0.587	0.491	0.514	0.547	0.531	0.676	0.722	0.656	1.000	
WF21	0.831	0.750	0.698	0.640	0.675	0.712	0.627	0.724	0.663	0.600	0.685	0.644	0.599	0.579	0.594	0.597	0.725	0.762	0.686	0.938	1.000

**Table 3 sensors-20-00020-t003:** Summary of the wavelet features selected from the intersection of correlation scalograms for the disease identification.

Wavelet Features	Wavelength/nm	Scale	*R* ^2^	Significance of *t*-Test
WF02	471	4	0.539	***
WF06	696	1	0.602	***
WF09	841	4	0.441	***
WF11	963	3	0.548	***
WF13	1069	3	0.422	***
WF21	2272	4	0.544	***

Note: *** indicates that the significance reaches 0.001 significant level.

**Table 4 sensors-20-00020-t004:** Feasibility of the wavelet features for identifying *Fusarium* head blight.

Validation	Field Truth
Wavelet Features	Healthy	*Fusarium* Head Blight	Sum	UA/%	OA/%	Kappa Coefficient
Six wavelet features in the whole spectral wavelength range	Healthy	31	3	34	91.2	88.7	0.775
*Fusarium* head blight	5	32	37	86.5
Sum	36	35	71	
PA/%	86.1	91.4		
Four wavelet features concentrated in the range of 400–1000 nm	Healthy	29	3	32	90.6	85.9	0.719
*Fusarium* head blight	7	32	39	82.1
Sum	36	35	71	
PA/%	80.6	91.4

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
