# Peer review of "Identification of Fusarium Head Blight in Winter Wheat Ears Using Continuous Wavelet Analysis"

_sensors, 2019, doi:10.3390/s20010020_

Round 1

Reviewer 1 Report

Please update the citation on the text by changing the “et al.” to an italic et al.  Rephrase Line 27~30, as it is incoherent. What is “reserved” means in line 31? Line 33, If the results identified unique features, what is the importance of “low redundancy”? Line 36~37, Normally, training data are not used to validate your model unless you have minimal data. It does not provide a good model result as you use the same data on your validation. Line 70, Please explain more on what do you mean with the non-imaging technique? Line 73, replace “These” with “Their” Line 75, This is not a very good reason of why you explore an upper-band when the lower band (less than 1000nm) already provided a good result. So why is it important to use a band higher than 1000? What is the cultivar under study? What is the reason for Experiment 2 collecting on a different locations other than it's done in 2019? Why not use the same locations on Experiment 1 and 2 in the same three locations? All data collection was done in the field? How did you maintain the distance between the sensor and the wheat? With this field of view, how did you constraint that the area of the object of interest is the only one that is on the field of view and not other parts of the field (soil, etc.)? Please add the manufacturer of the calibration panel or state if the panel comes from the spectrometer. Line 148~150, Please explain in detail what you mean with this statement? Please provide a detail explanation of the selection of this value? In Figure 2.Why not use the same plant in showing each side, it is clear that Fig. 2 has two different plants. Line 159, replace “Of them,” with “Where” Why not use the same disease severity groupings with the National Rules for Monitoring and Forecast; 0% healthy, and 1~100% for Fusarium head blight infected. What is the experimental design used? The authors used three different locations for Experiment 1, but the numbers of data are quite small. Can you provide how many were collected in location 1, 2, and 3 for Experiment 1? Line 168, The paper argument then of using band beyond 1000 nm becomes useless. Anything on SWIR is a water-related band - but why these 3 groups of SWIR were selected not to be used, needs have a better argument rather than just simply “strong absorption”. Since the data have been collected already - why not present all the data from 300~2500 nm. Line 173~174, What is the use of the calibration panel then? As presented in this paper that Experiment 1 data was used as training, and Experiment 2 was used for validation; what will this ratio impact the result of the validation? Noise from two locations are not related, and if it is, then you can always find a solution to noise minimization. In the case of atmospheric conditions as indicated by the authors, why not use the atmospheric condition of each location to minimized noise from each location if it was really the case? What is the importance of localizing the signal in the time-frequency domain? Is the occurrence of certain frequency in time relevant in this study? Line 204, Please show in a figure of the shape of the vegetation absorption as similar to Gaussian or quasi gaussian? What is the head blight infection ratio? Line 225, delete “selecting of” Please define entropy reduction? Line 245, replace “testing” with “validation” Figure 4a showed that there is a separation between ~900 ~ 1300. Add another figure that showed clearly the bands in Figure 4a as mentioned in this statement. Line 278~279, Why is there a need to do SWIR when the disease can be easily detected in the visible region? If the results enhanced the NIR region, then the visible or SWIR were also enhanced. But the most important aspect, is an enhancement of the separation really important? Line 341~342, this needs to be presented in the materials and methods. Line 345, How did the authors collect the data in the field by measuring the side of the infected ear with high accuracy? This was not presented in Materials and Methods. Line 346, What is “large machine”? Line 349, Time consuming? But there is no information on how fast this method performs? of even a comparison to other researches that have the same positive result? Please present the processing time of this method. The training accuracy is not really a good gauge as it is using training data. The high percentage is expected all the time, unless if the model is really bad.

Author Response

Dear Reviewer,

We really appreciate your suggestions and comments. We agree with these suggestions and have significantly revised the manuscript accordingly.

Our point-by point response and/ or changes to the reviewers’ suggestions/ comments are listed as follows, and the changes are marked in red.

Reviewer 2 Report

ear Authors,
Although your manuscript is, in general, an interesting contribution for the remote sensing community, there are some issues the author may need to address in the revision.

2.2.1 Wheat Ear Spectra Measurement
#1
Were the samples transported to the laboratory after cutting wheat plants?

#2
Could you clarify the position of the sensor?

#3
Were there any drifts depended on inherent variation in detector sensitivities? Did you apply the splice correction function to modify these connections?

#4 L.214
Could you define 'Fusarium head blight infection ratio'? How did you measure? Visual observation? Could you tell us its significant digit?

#5 LL.261-264
Did you copy these sentences from Kadariya (2006)? Why didn't you cite this literature?

Author Response

(The authors gave the same response as above.)

Round 2

Reviewer 1 Report

I want to thank the authors for their timely response. Most of the responses were acceptable except for a few important areas, which I think will make the manuscript better. Below are the important topics in the manuscript that needs to be addressed:

Minor:

Line 70. Delete “non-imaging hyperspectral data from”

Major:

8. In the previous manuscript in Line 75-77, the authors mentioned that other studies made on the same disease used 400 ~ 1,000 nm range. And so the problem on No. 8 was – the argument is very weak. I completely agree with the authors on why wavelength beyond the 1000 nm is needed IF there are problems with the lower bands. Unfortunately, that argument is too general and may not apply to winter wheat. Present another research, or cite a research article about winter wheat where the wavelengths determined for two or more different diseases are the same? The Number 9 response is inadequate. We need facts as much as possible in our responsibility to present our work to other researchers. So when the authors said - " we believed..." then it becomes an opinion. Please cite reference/s that different varieties do not matter in winter wheat when collecting data from wheat ears. Please add another image in Figure 1, where it showed a picture of the data collection in the field or the setup of the data collection in the field. For No. 17 - I think the authors - misunderstood my question. What I am referring to is - what is the experimental design or design of experiments (quasi-experimental, randomized block, split-plot, etc.) used for this work? For No. 19 - The recommendation is, why not show all the data and mark that area where the authors mentioned have noise. You have all the data. Just show all those in the figure and properly marked it. For No. 20. The authors need to address the following: Why did you divide the data from two other experiments when your calibration panel has to compensate for the changes of illumination. Normalization is the result of the calibration panel. There is no merit on why you divide your two data from two experiments. Please justify. For No. 21. This question is related to Number 20. Your explanation does not make sense. Again, the calibration panel is used to normalize your data. Dividing both so one will have more of the same ranges - is, unfortunately, unacceptable or does not make sense at all. If you have this kind of problem, the main responsibility of the authors is to make sure that your data are valid and so cross-reference and other best practices must be done. Rather than explain the inadequacy (human factor) and published. As responsible researchers, we always adhere to proper data collections, even if it takes another year. For No. 22. Response on 22 is inadequate, and the reviewer's comments are similar to 21. For No. 23. The response is not quite related to the question asked. Please response on what is the importance of localizing the signal in the time-frequency domain? Is the occurrence of certain frequency in time relevant in this study? For No. 24. Is the plot shown in the response of 24 a gaussian or a sinc function or mexican hat? For No. 29. The answer to this will depend on the answer in 8.

Author Response

(The authors gave the same response as above.)

Reviewer 2 Report

The authors added some elements to enrich manuscript and I think this paper can now be accepted for publication.

Author Response

Dear Reviewer, We really appreciate your suggestions and comments.

Round 3

Reviewer 1 Report

I am satisfied with the second response of the authors and the changes made from the original manuscript. The manuscript showed some big change as compared to its first version. Although the experimental design wasn't addressed properly, the authors' presentation on how they collected the data will suffice.